# Maternal Immune Activation Induced by Prenatal Lipopolysaccharide Exposure Leads to Long-Lasting Autistic-like Social, Cognitive and Immune Alterations in Male Wistar Rats

**DOI:** 10.3390/ijms24043920

**Published:** 2023-02-15

**Authors:** Emilia Carbone, Valeria Buzzelli, Antonia Manduca, Stefano Leone, Alessandro Rava, Viviana Trezza

**Affiliations:** 1Section of Biomedical Sciences and Technologies, Department of Science, Roma Tre University, Viale Guglielmo Marconi 446, 00146 Rome, Italy; 2Department of Physiology and Pharmacology, Sapienza University of Rome, Piazzale Aldo Moro 5, 00185 Rome, Italy; 3Neuroendocrinology, Metabolism and Neuropharmacology Unit, IRCCS Fondazione Santa Lucia, Via del Fosso di Fiorano 64, 00143 Rome, Italy; 4Department of Anatomical, Histological, Forensic Medicine and Orthopedic Sciences, Sapienza University of Rome, Via Alfonso Borelli 50, 00161 Rome, Italy

**Keywords:** neurodevelopmental disorders, MIA, cytokines, ASD, lipopolysaccharide, immune system, NDD, rat, inflammation

## Abstract

Several studies have supported the association between maternal immune activation (MIA) caused by exposure to pathogens or inflammation during critical periods of gestation and an increased susceptibility to the development of various psychiatric and neurological disorders, including autism and other neurodevelopmental disorders (NDDs), in the offspring. In the present work, we aimed to provide extensive characterization of the short- and long-term consequences of MIA in the offspring, both at the behavioral and immunological level. To this end, we exposed Wistar rat dams to Lipopolysaccharide and tested the infant, adolescent and adult offspring across several behavioral domains relevant to human psychopathological traits. Furthermore, we also measured plasmatic inflammatory markers both at adolescence and adulthood. Our results support the hypothesis of a deleterious impact of MIA on the neurobehavioral development of the offspring: we found deficits in the communicative, social and cognitive domains, together with stereotypic-like behaviors and an altered inflammatory profile at the systemic level. Although the precise mechanisms underlying the role of neuroinflammatory states in neurodevelopment need to be clarified, this study contributes to a better understanding of the impact of MIA on the risk of developing behavioral deficits and psychiatric illness in the offspring.

## 1. Introduction

In the last few years, there has been a growing interest in neurodevelopmental disorders (NDDs) related to maternal infections and associated inflammation. Indeed, increasing epidemiological and preclinical studies have provided substantial support to the association between maternal immune activation (MIA) during gestation and an increased susceptibility in the offspring to develop various psychiatric and neurological disorders, including (but not limited to) autism spectrum disorder (ASD) [1,2,3,4,5]. Specifically, several studies explored the correlation between intrauterine infections and altered fetal brain development, resulting in long-term cognitive and behavioral impairments due to the fetal inflammatory response, regardless of the specific pathogen involved [6]. In this context, a marked correlation between offspring immune activation and different cytoarchitectural alterations, leading to the onset of neuropsychiatric disorders such as schizophrenia, bipolar disorder and ASD, has been established [7,8,9,10].

The epidemiological data collected after many epidemics since the second half of the 20th century, including the evaluation of the potential detrimental outcomes of the recent COVID-19 pandemic, suggest MIA plays a role in the development of ASD [10,11,12], as confirmed by the chronic inflammation of the central nervous system (CNS) observed in patients with autism [13,14].

In preclinical studies, MIA is primarily modeled by the administration during early and/or mid-gestation of viral- or microbial-derived immunogens such as polyinosinic:polycytidylic acid (poly(I:C)) [15] and lipopolysaccharide (LPS) [16], respectively. Systemic endotoxin administration, such as LPS, in pregnant dams is sufficient to induce a significant pro-inflammatory response [17], as well as bacterial or viral molecules, which can cross the mother–fetus interface and affect the offspring’s immunity, metabolism and brain developmental trajectories, favoring the onset of several NDD-like behavioral disorders [18,19,20]. Interestingly, LPS—which is a component of the outer membrane of Gram-negative bacteria—can promote a pro-inflammatory cascade by binding to Toll-like receptor 4 (TLR-4), stimulating the innate immune response mainly through activated microglial/macrophages cells [21]. Thus, it evokes a series of alterations at the inflammatory and behavioral level, mainly due to the negative effect exerted on formation of mature synapses [9]. This evidence has been reported in many studies, in which prenatal exposure to LPS produced immune [22,23] and behavioral dysfunctions such as communicative deficits, social impairments and stereotypies in rodents [24,25,26,27]. These findings are also corroborated by growing evidence demonstrating the role of neuropoietic cytokines in neurodevelopment, including glio- and neurogenesis [28,29], fate determination, cell death, neuronal–glia crosstalk and synaptic connectivity [2,30,31,32]. In animal models, interruption of these neurodevelopmental processes during key stages of CNS maturation contributes to neurochemical and cytoarchitectural alterations in specific brain regions, such as the hippocampus and amygdala [19,20], suggesting that aberrant activation of the immune system in early life has important implications for brain development and related neurobiological functions in later life. Interestingly, ASD patients showed altered cytokine levels in the peripheral blood [33,34], strengthening the hypothesis of a correlation with behavioral alterations (i.e., altered social behavior) [14] and pointing to the fetal (immune) dysregulated pathways as a causative factor increasing vulnerability to NDDs in the offspring.

Despite this evidence, the paucity of data regarding the impact of maternal inflammation during pregnancy on offspring psychopathology and the intrinsic limitation of the few clinical studies available reinforce the importance of preclinical investigations in providing the data necessary to assess the etiologic and risk factors of NDDs, including ASD, with the purpose of defining effective countermeasures for patients and new biomarkers for early diagnosis.

In this context, the aim of the present study was to provide extensive characterization of the short- and long-term behavioral and immunological consequences of maternal exposure to LPS, mimicking a bacterial infection during gestation, in rat offspring across development. Specifically, we first evaluated the short- and long-term consequences of maternal exposure to LPS across several behavioral domains relevant to human psychopathological traits, including emotionality, sociability, stereotypic-like behaviors and cognitive deficits at different developmental ages (infancy, adolescence and adulthood) in the male rat offspring.

Moreover, inflammatory markers were measured in plasma samples of offspring across development to evaluate whether abnormal peripheral cytokines profiles were associated with behavioral deficits in LPS-treated rats both in adolescence and adulthood, and whether they represent potential biomarkers for environmentally triggered ASD.

## 2. Results

### 2.1. Reproduction Data

Prenatal LPS exposure did not affect gestation length, litter size at birth, male/female ratio, pups’ weight gain at different developmental ages and their postnatal viability. However, we observed a trend of LPS treated dams being unable to carry the pregnancy to term (t = 2.312, *p* = 0.08, df = 4, Table 1). We evaluated whether LPS injection induced maternal sickness behavior by measuring the dam’s weight from GD 0 to GD 11: thus, we weighed the dams at different time points before (at GD 0, GD 3, and GD 9) and after (at GD 10 and GD 11) LPS injection, to assess its potential impact on the dam weight gain immediately after LPS administration. We found that pregnant rats treated with LPS showed, at GD 10, a significant loss of weight compared to the day before (t = 5.445, *p* < 0.001, df = 7; Appendix A), probably due to a sickness behavior or, in some cases, abortion, which would support the trend towards reduced successful deliveries in LPS-treated dams.

### 2.2. Prenatal Exposure to LPS Caused Deficits in Social Communication in Infant Rat Offspring, While Maternal Care and Pup Homing Behavior Were Unaffected

LPS-exposed male offspring emitted less USVs at infancy when separated from the dam and siblings compared to male SAL-pups (PND 5: t = 2.272, *p* < 0.05, df = 27, Figure 1A; PND 9: t = 2.297, *p* < 0.05, df = 27, Figure 1B). These results suggest that prenatal exposure to LPS causes deficits in early social communication of the offspring, in line with the altered early communication profiles observed in both environmental [35,36,37] and genetic [38,39] animal models of ASD. The altered communication abilities displayed by LPS-exposed pups did not result from changes in maternal behaviors: indeed, in line with previous studies [40], prenatally LPS-exposed pups received the same maternal care as control pups (t = 0.509, *p* = n.s., df = 15, Figure 1C).

During the homing behavior test at PND 13, male pups prenatally exposed to LPS did not differ from control animals in their latency to reach the nest arena (t = 0.275, *p* = n.s., df = 27; Figure 1D), in the total time spent in the nest zone (t = 0.615, *p* = n.s., df = 27; Figure 1E) or in the number of entries in the nest zone (t = 0.733, *p* = n.s., df = 27; Figure 1F), suggesting that there were no alterations in the homing behavior as early indicator of social discrimination.

### 2.3. Prenatal Exposure to LPS Caused Deficits in Social Play Behavior and Sociability, but No Stereotypic Behaviors, in the Adolescent Offspring

To assess whether LPS-exposed adolescent animals showed impairments in the social domain, we tested the offspring using the social play behavior and three-chamber tests at PND 35. We found that adolescent rats prenatally exposed to LPS showed reduced social play behavior. In particular, although LPS- and SAL-exposed rats did not significantly differ in the frequency of pouncing (i.e., play solicitation; t = 1.612, *p* = n.s., df = 13; Figure 2A), rats prenatally exposed to LPS showed a significantly lower number of pinnings (t = 2.289, *p* < 0.05, df = 13; Figure 2B) and reduced play responsiveness (i.e., the percentage of response to play solicitation, calculated as the probability of an animal being pinned in response to play solicitation (pouncing) by the stimulus partner [41]; t = 2.628, *p* = n.s., df = 13; Figure 2C) compared to the control group. Prenatal LPS exposure selectively reduced social play behavior, as the total time spent in social exploration was unaffected (t = 0.676, *p* = n.s., df = 13; Figure 2D). At the same developmental age, LPS- and SAL-exposed animals were also tested in the three-chamber test. The results of this test revealed a general reduced sociability of LPS-exposed rats compared to the SAL-exposed animals (time sniffing stimulus: t = 3.733, *p* < 0.01, df = 15; Figure 2E; % sociability index: t = 4.333, *p* < 0.001, df = 15; Figure 2F). 

Conversely, LPS-exposed rats did not show repetitive behaviors compared to the SAL-exposed animals in the hole board test (Number of Head dippings: t = 0.790, *p* = n.s., df = 16, Figure 2G; number of rearings: t = 0.862, *p* = n.s., df = 16, Figure 2H).

### 2.4. Prenatal Exposure to LPS Induced Deficits in Social Discrimination and Sociability Together with Stereotypic-like Behaviors in the Adult Offspring, with No Changes in the Elevated Plus-Maze Test

The social discrimination test performed at PND 75 revealed an impaired discrimination ability in LPS-exposed animals: these animals, compared to SAL-exposed controls, failed to discriminate a familiar from an unfamiliar stimulus animal (% discrimination index: t = 2.722, *p* < 0.05, df = 17; Figure 3A); % time sniffing new (t = 2.722, *p* < 0.05, df = 17; Figure 3B).

Next, we tested the animals in the three-chamber test to evaluate whether the deficit in sociability displayed by LPS-exposed adolescent rats persisted into adulthood. We found that LPS-exposed adult rats spent less time sniffing the stimulus animal (t = 3.690, *p* < 0.01, df = 15; Figure 3C) and showed reduced sociability (% Sociability index: t = 3.851, *p* < 0.01, df = 15; Figure 3D) compared to the control group, indicating that prenatal LPS exposure induces an enduring deficit in sociability in the rat offspring.

When tested in the hole board test, adult rats prenatally exposed to LPS showed a significant increase in the frequency of head dippings compared to the control group (t = 2.192, *p* < 0.05, df = 16; Figure 3E), with no changes in the frequency of rearings (t = 1.481, *p* = n.s., df = 16; Figure 3F). 

We also performed an elevated plus-maze test to investigate whether prenatal LPS administration induced anxiety-related behaviors in the adult offspring. The results of this test showed no differences between the two experimental groups (% time open arms: t = 0.657, *p* = n.s., df = 15, Figure 3G; % open entries: t = 0.828, *p* = n.s., df = 15, Figure 3H).

### 2.5. Flow Cytometric Analysis

We performed a flow cytometric analysis on plasma samples to compare the peripheral inflammatory conditions between rats prenatally exposed to either LPS or saline, both at adolescence (Figure 4G) and adulthood (Figure 4H). At adolescence (PND 35), the flow cytometric analysis did not reveal any difference between plasma samples from LPS- and SAL-exposed animals for any of the 17 pro-/anti-inflammatory cytokines, particularly for IFN-γ (t = 0.652, *p* = n.s., df = 3, Figure 4A), MCP-1 (t = 1.918, *p* = n.s., df = 3, Figure 4B) and IL-13 (t = 1.223, *p* = n.s., df = 4, Figure 4C). Conversely, at adulthood, we found a statistically significant increase in the MFI values of IFN-γ (t = 2.840, *p* < 0.05, df = 7, Figure 4D), MCP-1 (t = 8.639, *p* < 0.001, df = 8, Figure 4E) and IL-13 (t = 4.047, *p* < 0.01, df = 7, Figure 4F) in LPS-exposed animals compared to control rats. These results reveal a compromised immune condition in the adult offspring prenatally exposed to LPS.

## 3. Discussion

The results of the present study show that maternal exposure to LPS causes long-lasting behavioral changes in the progeny beginning in early developmental periods, and also results in an altered cytokine profile in adulthood. In particular, we found that prenatal exposure to LPS affected isolation-induced USVs but not homing behavior in the male infant rat offspring, altered different facets of social behavior both at adolescence and adulthood, and induced stereotypic behaviors in the hole board test at adulthood only. Changes in cytokines were also detected at the systemic level at adulthood but not adolescence.

A significant body of evidence supports the multifactorial etiology of NDDs, including ASD: beyond genetic contribution, maternal exposure to environmental risk factors—e.g., drugs, malnutrition, inflammations/infections—has been linked to an increased risk of NDDs in the offspring. However, the mechanisms by which maternal environmental risk factors interfere with typical brain developmental trajectories, thereby influencing offspring vulnerability to NDDs, remain elusive. In the face of the substantial individual burden and the societal costs NDDs incur on public healthcare, it is urgent to mitigate/prevent the risk and severity of these conditions by guiding programs and policy decisions at global levels. In this context, preclinical research is essential for revealing the neurobiological mechanisms that mediate the offspring vulnerability to the detrimental effects of maternal environmental risk factors, including inflammation.

Here, we investigated the autistic-like behavioral and immunological consequences of a maternal bacterial infection, induced by LPS injection, in the male rat offspring across development. Based on literature data, we considered embryonic days 8–10 as a representative time interval corresponding to the first trimester of pregnancy in humans, when organogenesis occurs and microglial progenitors migrate to the CNS [9,42,43,44], making it a critical time window to study the impact of a bacterial-like maternal infection. Therefore, we administered LPS to rat dams at GD 9.5 and, in line with previous findings [22,26,35], we found autistic-like traits across development in animals exposed in utero to LPS. Interestingly, although reproduction parameters (e.g., gestation length, litter size at birth, male/female ratio, pups’ weight gain at different developmental ages and postnatal viability) were unaffected by maternal exposure to LPS, all treated dams displayed a marked weight reduction the day after LPS exposure. This suggests a temporary sickness condition that may account for the reduced trend in successful deliveries displayed by LPS-treated dams. Although this association did not reach statistical significance, the observed trend could be interpreted as a miscarriage event, and therefore future experiments should reconsider the choice of GD of administration, LPS serotype or dose.

Our results revealed long-lasting behavioral alterations in LPS-exposed animals, which are apparently independent of the maternal care received during infancy, as both dams treated with SAL and LPS exhibited an equal number of maternal behaviors. In infancy, LPS-exposed pups showed a communicative deficit, as they emitted a lower number of USVs when separated from the mother and littermates within the first two weeks of life (i.e., at both PND 5 and 9). Infant rodents produce USVs in response to separation from the mother and the nest. These USVs play an important role in mother–offspring interactions. They are an indicator of emotional reactivity in the pups and their measurement represents a potent tool used to detect subtle effects of adverse events during development [45].

During the early phases of postnatal life, olfaction—and in particular the learned association between maternal odors and maternal stimulation—is crucial to the development of social behavior and social recognition [46]. Therefore, we tested the infant offspring in the homing behavior test, which requires intact sensory, olfactory and motor capabilities that allow the pup to recognize the mother’s odor among others [47,48,49]. We did not find any difference between the two experimental groups in this task, suggesting that intact olfactory-based discriminative abilities are present in LPS-exposed pups at this early stage of development.

During development, we assessed the impact of prenatal LPS exposure on different facets of the social repertoire of the offspring, using specific tasks: (1) a free dyadic social encounter procedure with a same-age stimulus animal at adolescence, which we used to study social play behavior (that is, the most characteristic form of social behavior displayed by rodents between weaning and sexual maturation); (2) the three-chamber test, performed both at adolescence and adulthood, which focuses on the social approach of the experimental animal toward a confined stimulus animal, without direct social contact; (3) the social discrimination test, which is designed to assess the discriminative abilities of rodents on the basis on their innate preference for exploring a new rather than a familiar social partner.

In line with previous studies [50,51], prenatal LPS exposure reduced social play behavior in the adolescent offspring, as LPS-exposed rats showed reduced pinning and play responsiveness compared to SAL-exposed animals. Interestingly, these changes were restricted to play-related behaviors, as general social interaction during the test session was unaffected. At the same developmental age, LPS-exposed animals also showed impaired sociability in the three-chamber test, as they spent less time in social approach with the stimulus animal compared to control animals. Notably, deficits in different components of social behavior were also observed in adulthood: thus, adult LPS-exposed rats showed impaired sociability (in the three-chamber test) and altered social discrimination abilities (in the social discrimination test). These data highlight that prenatal LPS exposure induces in the rat offspring a wide range of impairments in the social domain beginning in early developmental stages, in line with clinical observations that children, adolescents and adults with ASD demonstrate marked socio-communicative deficits [52,53,54].

Together with deficits in social and communicative domains, the hallmark characteristics of ASD include repetitive and/or stereotyped behaviors. To evaluate possible stereotypic-like behaviors following prenatal exposure to LPS, we tested the adolescent and adult offspring in the hole board test. In line with recent findings reporting that adult LPS-exposed animals showed increased repetitive self-grooming behavior and stereotypies in an open field arena [26], we found that LPS-exposed animals showed enhanced stereotypic/repetitive behaviors only in adulthood, as the number of head dippings increased compared to controls. To evaluate whether this behavior could be related to an anxious phenotype, we also performed an elevated plus maze test: however, adult LPS-exposed rats did not display anxiety-like behaviors in this task, as the percentage of time spent in the open arms and the number of open arms entries did not significantly differ between SAL- and LPS-exposed rats.

Overall, we have provided extensive characterization of the short- and long-term behavioral consequences of MIA induced by prenatal LPS exposure in the rat offspring across development, showing deficits in the communicative, social and cognitive domains, together with stereotypic-like behaviors in LPS-exposed animals.

To evaluate whether changes in cytokine levels were present in these cohorts of animals, we compared the peripheral inflammatory conditions between rats prenatally exposed to either LPS or SAL, both in adolescence and adulthood. Interestingly, among the 17 pro-/anti-inflammatory cytokines analyzed, we found increased levels of IFN-γ, MCP-1 and IL-13 in the plasma of adult LPS-exposed animals only, highlighting a discrete alteration in their peripheral inflammatory profile. Generally, higher levels of these cytokines have been associated with a diagnosis of ASD [55,56,57], supporting a mechanistic link between gestational inflammation and development of ASD-like characteristics in the progeny.

These cytokines coordinate a variety of immune system responses, acting as key signaling molecules that regulate processes involved in induction/resolution of inflammation [58]: for instance, among the multitude of functions covered by IFN-γ are the regulation of antigen presentation, the promotion of inflammatory and chemotactic signals, as well as the antagonization of suppressive cytokine expression [59,60]. Interestingly, preclinical evidence showed that the absence of IFN-γ in the dorsal hippocampus of adult mice ameliorates synaptic plasticity and cognitive performance [61]. This seems to correlate with our results showing that adult LPS-exposed animals had increased plasmatic levels of IFN-γ compared to SAL-treated controls, together with reduced performance in the social discrimination task.

In addition to IFN-γ, we also found increased levels of MCP-1 (the Monocyte chemoattractant protein-1, also known as Chemokine CC-motif ligand 2 (CCL2)) in adult LPS-exposed animals. This chemokine is crucial in promoting inflammation through macrophages activation, monocyte recruitment and modulation of several other pro-inflammatory factors [62]. Moreover, MCP-1 is also considered a biomarker of chronic inflammation in Alzheimer’s disease [63] and its expression has been associated with increased oxidative stress production, suggesting the pivotal role this chemokine plays in the inflammatory processes, as well as in several pathological conditions [62]. In line with our findings, increased levels of various chemokines including (but not limited to) MCP-1 have been reported in the brain, as well as in peripheral blood from children with ASD, and they have been linked to their deterioration in communication skills, stereotyped behaviors and hyperactivity, and to their impairment of cognitive functions and adaptive skills [64,65,66]. Finally, we found that adult LPS-exposed animals also showed increased levels of IL-13, an anti-inflammatory cytokine which is well known for its neuroprotective activity, mainly exerted by the modulation of microglial alternative states [9,67,68]. Together with our present findings, recent evidence in humans also reported that individuals with ASD showed significantly higher concentrations of IL-13 in comparison to their matched controls [56], suggesting the potential of IL-13 as a biomarker in the diagnosis of ASD.

Although we did anticipate age-related changes in the peripheral inflammatory profile of LPS-exposed animals (in line with the observed detrimental behavioral profile across development), adolescent LPS-exposed animals did not show significant differences in any of the 17 pro-/anti-inflammatory cytokines analyzed, including IFN-γ, MCP-1 and IL-13. However, it is still possible that changes in cytokine levels may occur in specific brain regions of adolescent LPS-exposed animals. There is indeed evidence that peripheral inflammation (such as that induced by LPS) causes a “mirror” inflammatory response in the CNS, which is characterized by additional synthesis and action of cytokines within the brain [69]. Although the precise mechanisms responsible for CNS synthesis of cytokines are not entirely understood, the cytokines ultimately produced in the brain can have several sources, including microglia, invading inflammatory cells, microvessel endothelial cells, astrocytes and even neurons in which cytokines can be constitutively expressed [70]. In our opinion, this remains a fascinating hypothesis to investigate in a follow-up of this study and underscores the importance of research focusing on the exact role of different cytokines in brain–periphery interactions during normal (fetal) development, and in NDDs such as ASD. Since sex-dependent differences in preclinical models of ASD have been documented [71,72,73], together with sexual dimorphism in a neuroinflammatory profile following perinatal inflammation [74], the inclusion of both male and female animals should be considered in future studies.

Taken together, the current findings indicate that MIA induced by prenatal LPS exposure leads to long-lasting social, cognitive and immune alterations in male Wistar rats. However, further investigations are required to dissect the mechanisms underlying the contribution of inflammatory immune activity to disrupted neurobehavioral function in the progeny, including females.

## 4. Materials and Methods

### 4.1. Animals

Female Wistar rats (Charles River, Calco, Italy), weighing 250 ± 15 g, were mated overnight. The morning when spermatozoa were found was designated as gestational day (GD) 0 [22,35,50]. Pregnant rats were singly housed in Macrolon cages (40 cm (length) × 26 cm (width) × 20 cm (height) cm), under controlled conditions (temperature 20–21 °C, 55–65% relative humidity and 12/12 h light cycle with lights on at 07:00 a.m.). Food and water were available ad libitum. On GD 9.5, females received a single intraperitoneal (i.p.) injection of either LPS or saline (SAL) [22,26,35]. Newborn litters found up to 17.00 h were considered to be born on that day (postnatal day (PND) 0). On PND 1, the litters were culled to eight animals (six males and two females), to reduce the litter size-induced variability in the growth and development of pups during the postnatal period. On PND 21, the pups were weaned and housed in groups of three. The experiments were carried out on the male offspring during infancy (PNDs 5, 9 and 13), adolescence (PND 35) and early adulthood (PND 75). One pup per litter, from different litters per treatment group, was randomly used in each behavioral experiment and not re-used in subsequent experiments, except for flow cytometric analysis. For every experiment, the sample size (n) for each experimental group/condition is indicated in the figure legends. Scatter dot plots represent each animal (or a pair of animals for the social play experiments). The sample size was based on our previous experiments and power analysis (G*Power 3.1 software).

The experiments were performed in agreement with the ARRIVE (Animals in Research: Reporting In Vivo Experiments) guidelines [75], the guidelines of the Italian Ministry of Health (D.L. 26/14) and the European Community Directive 2010/63/EU and were approved by the Italian Ministry of Health (authorization n. 1207/2016-PR).

The timeline of the experiments is available in the Appendix A and the sample size for each experiment is indicated in the figure legends.

### 4.2. Drugs

LPS (from *Escherichia coli*, serotype O127:B8, Sigma Aldrich-Merck, Darmstadt, Germany) was dissolved in saline solution at a concentration of 100 µg/kg and administered to pregnant dams (i.p. in a volume of 2 mL/kg at GD 9.5). This protocol of administration has been shown to induce some autistic-like behavioral changes in the rat offspring [22,26,35].

### 4.3. Reproduction Data

Reproduction data were measured, including the length of gestation (in days), the % of successful deliveries (calculated as the number of successful deliveries/the number of total pregnancies × 100) [76], the litter size, the ratio male/female, the weight gains of pups and the % of postnatal vitality (calculated as the number of pups at PND 21/the number of pups at PND 1 × 100) [76].

Furthermore, pregnant rats were weighed regularly to evaluate the consequences of LPS injection on their body weight [50].

### 4.4. Maternal Behavior Observation

Maternal behavior was assessed daily in the colony room, from PND 2 to PND 13, by well-trained experimenters, who were blinded to experimental groups. The assessments occurred at regular intervals of 3 min in 3 sessions of 72 min each during the light phase (09:00 a.m., 01:00 p.m., 05:00 p.m.), as previously described by Colucci and colleagues [77]. Literature data indicate that maternal behavior is high during the light phase and declines during the dark phase of the light/dark cycle [78,79]. In fact, during the dark (active) phase, self-directed behaviors in mothers increase compared to pup-directed behaviors, such as licking/grooming behaviors [78]. During each session, each dam and its litter were observed every 3 min (25 observations per 3 sessions per day for a total of 75 observations per day). We measured 7 maternal parameters: (1) arched nursing (dam adopting a nursing posture with its back and ventral surface arched over its pups), (2) blanket nursing (dam over the pups in nursing posture but not arched), (3) passive nursing (dam adopting nursing posture lying either on its back or side), (4) licking pups (dam licking pups), (5) pup retrieval (dam moving the pups in another cage position), (6) building nest (dam manipulating nest shavings), (7) maternal self-grooming (dam grooming its breasts); and four non-maternal parameters: (1) feeding, (2) exploring (exploring the cage), (3) not-exploring without pups (dam away from the pups), (4) self-grooming (grooming its body but not the breast).

### 4.5. Isolation-Induced Ultrasonic Vocalizations

Isolation-induced ultrasonic vocalizations (USVs) are emitted by rodent pups when they are removed from the nest. These vocalizations play an important communicative role in mother–offspring interactions. The isolation-induced USVs emitted by SAL- and LPS-exposed pups were recorded as previously described [37,45] at PNDs 5 and 9. Briefly, pups were individually removed from the nest and placed into a black Plexiglass arena (30 cm × 30 cm), located inside a sound-attenuating and temperature-controlled chamber. Pup USVs were detected for 3 min by an ultrasound microphone (Avisoft Bioacoustics, Berlin, Germany) sensitive to frequencies between 10 and 200 kHz and fixed at 15 cm above the arena. They were analyzed quantitatively (number of calls/3 min).

### 4.6. Homing Behavior

The homing behavior test exploits the tendency of immature rodent pups to maintain body contact with the dam and siblings, and to discriminate their own home cage odor from a neutral odor, which is an early indicator of social discrimination [47]. The homing behavior test was performed as previously described [80]. Briefly, on PND 13, the litter was separated from the dam and kept for 30 min in a temperature-controlled holding cage. Then, each pup was placed into a Plexiglass box with 1/3 of its floor covered with bedding from the pup’s home cage and 2/3 of its floor covered with clean bedding. The pup was located at the side of the box covered by clean bedding and its behavior was video recorded for 4 min for subsequent analysis. The following parameters were scored by an observer, who was unaware of the animal treatment, using the Observer 3.0 software (Noldus Information Technology BV, Wageningen, The Netherlands): their latency (s) to reach the home-cage bedding area; total time (s) spent by the pup in the nest bedding area; total number of entries into the nest bedding area; and locomotor activity, expressed as the total number of crossings in the test box.

### 4.7. Social Play Behavior

Social play behavior is one of the earliest forms of non-mother-directed social behavior which is highly expressed by young mammals, including rats, between weaning and sexual maturation [81]. The test was performed at PND 35 in a sound-attenuated chamber under dim light conditions, as previously described [82]. The testing arena consisted of a Plexiglass cage (40 cm × 40 cm × 60 cm) with ~2 cm of wood shavings covering the floor. Rats were individually habituated to the test cage for 10 min on 2 days prior to testing. On the test day, rats were socially isolated for 3.5 h prior to testing. This isolation period has been shown to induce a half-maximal increase in the amount of social play behavior [83]. The test consisted of placing two animals into the test cage for 15 min. The animals in a test pair did not differ more than 10 g in body weight and had no previous common social experience (i.e., they were not cage mates). A pair of rats was considered as one experimental unit and the behavioral parameters were therefore scored per pair of animals. The Observer 3.0 software (Noldus Information Technology BV, Wageningen, The Netherlands) was used to score behaviors related to play. In rats, a bout of social play behavior starts with one rat soliciting (“pouncing” on) another animal by attempting to nose or rub the nape of its neck. If the animal that is pounced upon fully rotates to its dorsal surface, this results in “pinning” (one animal lying with its dorsal surface on the floor with the other animal standing over it). This is considered the most characteristic posture of social play behavior in rats [84].

We determined (a) the frequency of pinning, (b) the frequency of pouncing, (c) the time spent in social exploration (i.e., the total amount of time spent engaging in non-playful forms of social interaction, such as sniffing any part of the test partner’s body, including the anogenital area, or grooming any part of the partner body) and (d) the “play responsiveness” (that is, the percentage of response to play solicitation) as the probability of an animal of being pinned in response to play solicitation (pouncing) by the stimulus partner [36,41].

### 4.8. Three-Chamber Test

The test was performed as previously described [85]. The apparatus was a rectangular three-chamber box with two lateral chambers (30 cm × 35 cm × 35 cm; *l* × *w* × *h*) connected to a central chamber (15 cm × 35 cm × 35 cm; *l* × *w* × *h*). Each lateral chamber contained a small Plexiglass cylindrical cage. At PNDs 35 and 75, each experimental rat was individually allowed to explore a three-chamber apparatus for 10 min and then confined in the central compartment. An unfamiliar stimulus animal was confined in a cage located in one chamber of the apparatus, while the cage in the other chamber was left empty. Both doors to the side chambers were then opened, allowing the experimental animal to explore the apparatus for 10 min. The percentage of time spent in social approach (sniffing the stimulus animal) and the percentage of the ratio between the time spent in social approach and the time spent exploring the social chamber (% Sociability index) were scored using the Observer 3.0 software (Noldus Information Technology BV, Wageningen, The Netherlands). For both PND 35 and PND 75, we used different animals. 

### 4.9. Hole Board Test

The test was performed in a sound-attenuated chamber under dim light conditions, as previously described [36] at both PNDs 35 and 75 by using different animals. The apparatus consisted of a gray square metal table (40 cm ×40 cm × 10 cm; *l* × *w* × *h*) with 16 evenly spaced holes (4 cm in diameter), inserted in a Plexiglass arena (40 cm × 40 cm × cm 60 cm; *l* × *w* × *h*). Each rat was individually placed in the apparatus for 5 min. Each session was recorded with a camera positioned above the apparatus for subsequent behavioral analysis performed using the Observer 3.0 software (Noldus Information Technology BV, Wageningen, The Netherlands). Dipping behavior was scored by the number of times an animal inserted its head into a hole, at least up to the eye level, as a measure of stereotyped/repetitive behaviors. Spontaneous rearing behaviors, in which rodents stand on their hind legs with the intention of exploring, were also counted during the 5 min test period.

### 4.10. Social Discrimination Test

The test was performed at PND 75. Briefly, animals were isolated for 7 days before testing [86]. The test consisted of a learning trial and a retrieval trial, which were separated by a 30 min intertrial interval. During the learning trial, an adolescent (30 days old), unfamiliar rat was introduced into the home cage of the experimental rat for 5 min. The time the experimental rat spent investigating (i.e., sniffing, allogrooming and following) the adolescent was measured. Thirty minutes after, the adolescent used in the learning trial was returned to the same adult’s cage together with a novel adolescent. Each session was recorded with a camera positioned above the cage for subsequent behavioral analysis performed using the Observer 3.0 software (Noldus Information Technology BV, Wageningen, The Netherlands). The time the adult spent exploring the novel and the familiar adolescents was monitored for 5 min. The discrimination index was calculated as the difference in time exploring the novel versus the familiar animal, expressed as the percentage ratio of the total time spent exploring both the animals [86].

### 4.11. Elevated Plus-Maze Test

The apparatus comprised two open (50 × 10 × 40 cm^3^; *l* × *w* × *h*) and two closed (50 × 10 × 40 cm^3^; *l* × *w* × *h*) arms that extend from a common central platform (10 × 10 cm^2^). The test was performed as previously described [87]. Rats were individually placed on the central platform of the maze for 5 min. Each session was video-recorded for subsequent behavioral analysis, which was performed using the Observer 3.0 software (Noldus Information Technology, The Netherlands). Immediately after each session, the apparatus was thoroughly cleaned with cotton pads wetted with 70% ethanol–water solution and dried to eliminate residual odors. The following parameters were analyzed: (1) % Time spent in open arms (%TO): (seconds spent on the open arms of the maze/seconds spent on the open + closed arms) × 100; (2) % Open arm entries (% OE): (the number of entries into the open arms of the maze/number of entries into open + closed arms) × 100. The test was performed between 9 a.m. and 2 p.m. under low-light conditions (2 lux).

### 4.12. Flow Cytometric Analysis

Trunk blood samples were collected from both SAL- and LPS-exposed offspring at adolescence and adulthood (PNDs 35 and 75, respectively) for flow cytometric analysis, following behavioral experiments. Up to 5 mL of blood was collected from each animal into 15 mL tubes filled with EDTA solution 0.5 M, pH 8 (Invitrogen—Thermo Fisher Scientific, Waltham, MA, USA) at room temperature (concentration 1:10). A different funnel was used for each subject. The tubes were then centrifuged for 20 min at 1000× *g* and at a temperature of 4 °C, within a maximum of 30 min from collection, using the Eppendorf^®^ Centrifuge 5810 R. Subsequently, supernatant was collected and partitioned into two aliquots, then the plasma was stored at −80 °C. Blood sampling procedures were performed according both to literature data [88,89] and to our experience. Plasma samples were then analyzed through a multiplex assay to perform a screening on a panel of 17 key pro- and anti-inflammatory rat cytokines (FirePlex^®^-96 Key Cytokines (Rat) Immunoassay Panel—ab235662, Cambridge, UK; CytoFLEX—Beckman Coulter, Indianapolis, IN, USA). Next, 10,000 hydrogel particles were acquired for each sub-population. A proper gate was designed for each sub-population, which represents a specific cytokine, and a Mean Fluorescence Intensity (MFI) was calculated by a dedicated FirePlex^®^ Analysis Workbench software, which decodes FirePlex particles, plots standard curves and outputs results.

### 4.13. Statistical Analysis

Reproduction and behavioral data are expressed as mean ± S.E.M. (Standard Error of the Mean), whereas the flow cytometric analysis data are expressed as mean ± S.D. (Standard Deviation). To assess the impact of LPS exposure in the dams and offspring, data were analyzed using unpaired and paired Student’s *t*-tests. The accepted value for significance was set at *p* < 0.05. Data were analyzed through GraphPad Prism 8 software.

## Figures and Tables

**Figure 1 ijms-24-03920-f001:**
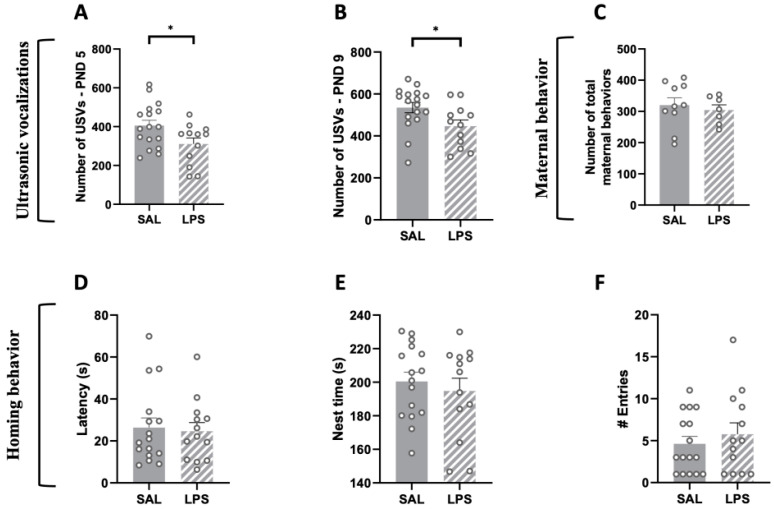
Impact of prenatal LPS exposure on maternal behavior, social communication and social discrimination in the infant rat offspring. LPS-exposed pups vocalized significantly less compared to SAL-exposed pups at both PNDs 5 (**A**) and 9 (**B**) (PND 5: SAL = 17, LPS = 12; PND 9: SAL = 17, LPS = 12). The altered USV profile displayed by LPS-exposed pups was not secondary to changes in maternal care (**C**) (Dams: SAL = 10, LPS = 7). LPS-exposed rats showed intact social discrimination abilities in the homing behavior task, as indicated by their latency to reach the nest arena (**D**), the total time spent in the nest zone (**E**) and the number of entries in the nest zone (**F**) (PND 13: SAL = 16; LPS = 13). Data represent mean ± S.E.M. * *p* < 0.05 vs. SAL (Student’s *t*-test).

**Figure 2 ijms-24-03920-f002:**
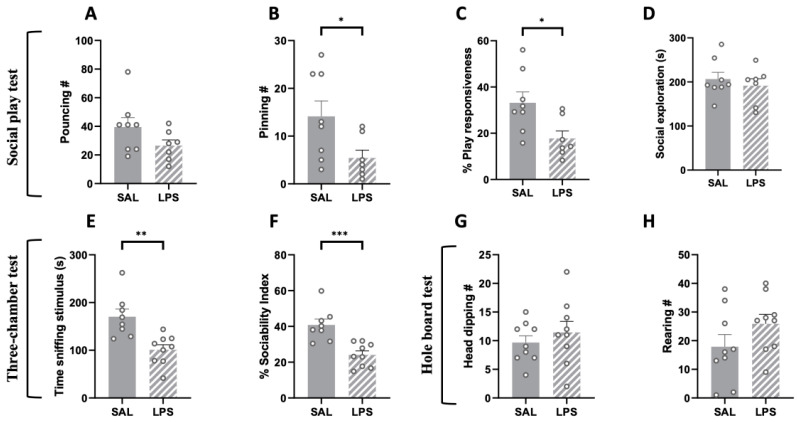
Effects of prenatal LPS exposure in the social play, three-chamber and hole board tests in the adolescent rat offspring. Social play behavior was reduced in the adolescent offspring prenatally exposed to LPS (**A**–**C**), while general social exploration was unaffected (**D**) (SAL = 8, LPS = 7). LPS-exposed adolescent rats also showed reduced sociability in the three-chamber test, as they spent less time sniffing the stimulus animal (**E**) and showed a reduced sociability index (**F**) in this task (SAL = 8, LPS = 9). No differences between the two experimental groups were found in the hole board test (number of head dipping (**G**); rearing (**H**)), indicating no stereotypic-like behaviors displayed by adolescent LPS-exposed rats in this task (PND 35: SAL = 9; LPS = 9). Data represent mean ± S.E.M. * *p* < 0.05, ** *p* < 0.01, *** *p* < 0.001 vs. SAL (Student’s *t*-test).

**Figure 3 ijms-24-03920-f003:**
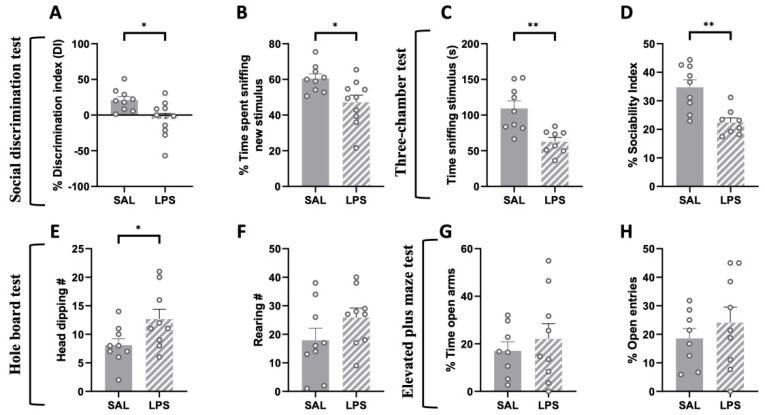
Effects of prenatal LPS exposure in the social discrimination, three-chamber, hole board and elevated plus-maze tests in the adult rat offspring. The social discrimination task revealed impaired social recognition abilities in adult LPS-exposed animals, since they displayed a lower discrimination index (**A**) and spent less time sniffing the new stimulus animal (**B**) compared to control animals (SAL = 9, LPS = 10). Adult LPS-exposed rats showed altered sociality in the three-chamber test (**C**,**D**) (SAL = 9, LPS = 8). These animals also showed an increased number of head dippings in the hole board task compared to the control group (**E**), with no changes in the frequency of rearing (**F**) (SAL = 9, LPS = 9). No differences between the two experimental groups were found in the elevated plus-maze test (% time open arms (**G**); % open entries (**H**)) (SAL = 8, LPS = 9). Data represent mean ± S.E.M. * *p* < 0.05, ** *p* < 0.01 vs. SAL (Student’s *t*-test).

**Figure 4 ijms-24-03920-f004:**
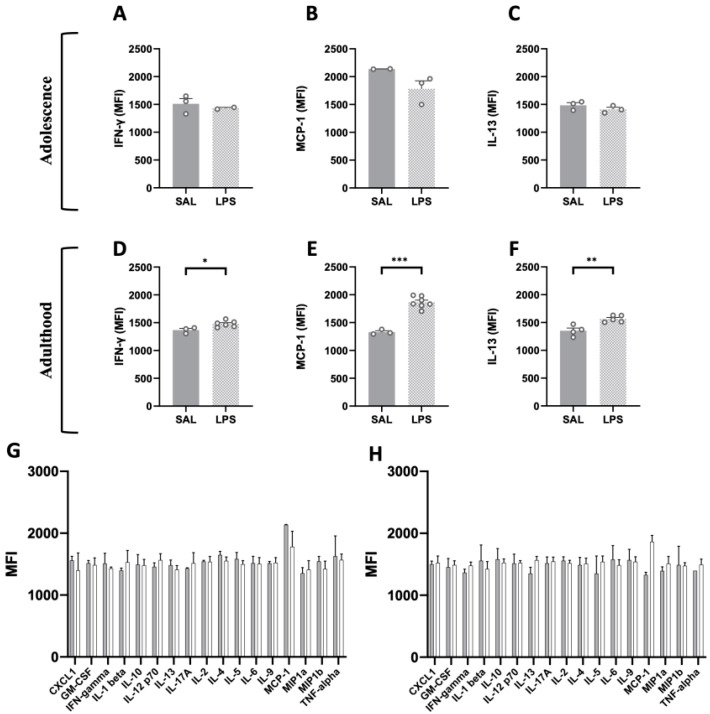
Flow cytometric analysis of pro- and anti-inflammatory cytokines in plasma samples of adolescent and adult LPS-exposed rats. The screening performed on a panel of 17 pro- and anti-inflammatory factors (FirePlex^®^-96 Key Cytokines (Rat) Immunoassay Panel) at adolescence (**G**) and adulthood (**H**) (PND 35: SAL = 2–7, LPS = 2–7; PND 75: SAL = 2–7, LPS = 2–7) revealed age-specific changes in three key cytokines: the MFI values of two pro-inflammatory cytokines, respectively IFN-γ (**A**) and MCP-1 (**B**), and the anti-inflammatory cytokine IL-13 (**C**) did not differ among LPS and SAL group in the adolescent progeny (PND 35: SAL = 2–3, LPS = 2–3), but their MFI values in the adult offspring were significantly increased compared to the control group (**D**–**F**) (PND 75: SAL = 3–4, LPS = 5–7). Data represent mean ± S.D. * *p* < 0.05, ** *p* < 0.01, *** *p* < 0.001 vs. SAL (Student’s *t*-test).

**Table 1 ijms-24-03920-t001:** Reproduction parameters. The table shows the length of gestation (in days), the % of successful deliveries, the litter size, the male/female ratio, the weight gain of pups and the % of postnatal vitality (Dams: SAL = 9, LPS = 8; Pups: SAL = 19; LPS = 17). Data represent mean ± S.E.M. (Student’s *t*-test): *p* = 0.08 vs. SAL.

Group	Gestation Length (Days)	Successful Deliveries (%)	Litter Size at Birth	Male/Female Ratio	Pups’ Weight				Postnatal Vitality (%)
					PND1	PND5	PND9	PND13	
SAL	22.24 ± 0.14	94.44 ± 5.56	14.47 ± 0.82	1.39 ± 0.26	7.42 ± 0.22	12.76 ± 0.20	22.83 ± 0.60	32.94 ± 0.53	100 ± 0
LPS	21.88 ± 0.12	60.65 ± 13.52	14.06 ± 0.75	1.17 ± 0.21	7.99 ± 0.33	13.33 ± 0.31	21.50 ± 0.54	33.21 ± 0.53	100 ± 0

## Data Availability

Data will be available upon request.

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
