# Peer review of "Maternal Immune Activation Induced by Prenatal Lipopolysaccharide Exposure Leads to Long-Lasting Autistic-like Social, Cognitive and Immune Alterations in Male Wistar Rats"

_ijms, 2023, doi:10.3390/ijms24043920_

Round 1

Reviewer 1 Report

What was the reason for expressing the variability of body weight data as S.D., instead of S.EM., as in the case of other parameters ?

I would advice You to recalculate the statistical significance level of body weight drop of dams one day after LPS-administration (the P<0.001 level, as denoted, seems to me to high for the given S.D.'s on the graph).

Were the pups choosen at random from the litter ? Could be the same pup used again ?

Author Response

Reviewer’s comment 1:

What was the reason for expressing the variability of body weight data as S.D., instead of S.EM., as in the case of other parameters? I would advice You to recalculate the statistical significance level of body weight drop of dams one day after LPS-administration (the P<0.001 level, as denoted, seems to me to high for the given S.D.'s on the graph).

Authors’ reply to Reviewer’s comment 1:

We thank the Reviewer for this comment. All the reproduction (including the weight of dams reported in S2 Figure) and behavioral data are expressed as mean ± S.E.M. Only flow cytometric analysis are expressed as mean ± S.D.

Reviewer’s comment 2:

Were the pups choosen at random from the litter ? Could be the same pup used again ?

Authors’ reply to Reviewer’s comment 2:

We thank the Reviewer for this comment. One pup per litter, from different litters per treatment group, was randomly used in each behavioral experiment and not re-used in subsequent behavioral experiments, in order to avoid the so-called “litter effect”. Concerning the flow cytometric experiments, blood samples were taken from animals randomly chosen from the animals performing the behavioral experiments at both adolescence and adulthood. This information has been now added in the revised manuscript.

Reviewer 2 Report

The manuscript by Carbone et al investigates the impact of LPS administration to pregnant rats on subsequent maternal and offspring behavioural parameters and on immune markers.  LPS administration to pregnant rodents is a standard model of maternal infection and such maternal immune activation (MIA) has been linked to increase risk of subsequent neuronal diseases in offspring. In particular MIA is linked to increase incidence of autism spectrum disorders and schizophrenia in offspring making these type of studies very important in determining underlying mechanisms and possible treatment strategies.

The results demonstrate some behavioural abnormalities in MIA offspring related to socialisation that are identified at adolescence and persist to adulthood. Some subtle alterations in cytokines are observed, although the functional significance and relationship (if any) to the behavioural abnormalities unclear  The rigour of the behavioural data and the thorough way the behavioural methods are described was excellent. I thought this was a real strength of the paper. No mechanistic insights are provided, although the findings provide a solid foundation for what I hope are subsequent studies that may try to probe mechanisms or ways to mitigate against the abnormalities.

I have a few minor suggestions the authors should consider.

1.     I feel it will be more informative for the reader to plot the data in Figs1-3 and 4A-F to show the individual data points overlaid on the mean +/- SEM bar graphs.

2.     Line 85 “and” to “an”

3.     Methods 2.11. Was the open arm maze (and other apparatus) were cleaned between tests to eliminate residual odours.

4.     3.4, line 377 and panel 3H. Is “time open entries” correct? (if so what is it?). Shouldn’t the parameters be relative time in open and closed arms, and also number of entries to the open arm?

5.     Discussion, line 420. Be careful about linking the behaviour and the cytokine levels as we don’t really know what the relationship (if any) is except they are observed in the same cohorts. You cant really say “behavioural changes were associated with..”. Perhaps something like “Changes in cytokines were also seen in these cohorts” is better.

6.     Lines 435-437. “ We evaluated several experimental protocols described in the literature, assessing the correlation between the severity of the external insult received, the gestational period, the corresponding brain developmental processes”. This should be deleted or rephrased. You didn’t assess severity of injury, gestaional insult period or brain development.

7.     Lines 449-451. Have the authors considered separating or excluding the offspring from individual animals that showed a drastic loss of weight. It may be a way to separate sickness from more subtle inflammatory challenge (although of course they are related so may not be sensible?).

8.     Line 469 Given the reduced USVs observed, is it really “normal….cognitive processing”?

9.     Lines 523-527 discusses IFN changes and how it impacts fetal development but you havnt measured fetal IFN, so parts of this not relevant. More relevant is whether adult increases in IFN (as you saw) cause cognitive abnormalities – eg could describe acute effects – or whether adult IFN changes reflect fetal changes.

10.  Lines 552-553. Very good point.

Author Response

Reviewer #2

Reviewer #2 (Comments to the Authors):

The manuscript by Carbone et al investigates the impact of LPS administration to pregnant rats on subsequent maternal and offspring behavioural parameters and on immune markers.  LPS administration to pregnant rodents is a standard model of maternal infection and such maternal immune activation (MIA) has been linked to increase risk of subsequent neuronal diseases in offspring. In particular MIA is linked to increase incidence of autism spectrum disorders and schizophrenia in offspring making these type of studies very important in determining underlying mechanisms and possible treatment strategies.

The results demonstrate some behavioural abnormalities in MIA offspring related to socialisation that are identified at adolescence and persist to adulthood. Some subtle alterations in cytokines are observed, although the functional significance and relationship (if any) to the behavioural abnormalities unclear  The rigour of the behavioural data and the thorough way the behavioural methods are described was excellent. I thought this was a real strength of the paper. No mechanistic insights are provided, although the findings provide a solid foundation for what I hope are subsequent studies that may try to probe mechanisms or ways to mitigate against the abnormalities. I have a few minor suggestions the authors should consider.

Reviewer’s comment 1:

I feel it will be more informative for the reader to plot the data in Figs1-3 and 4A-F to show the individual data points overlaid on the mean +/- SEM bar graphs.

Authors’ reply to Reviewer’s comment 1:

Following the Reviewer’s comment, individual datapoints are now reported in the revised Figures of the manuscript.

Reviewer’s comment 2:

Line 85 “and” to “an”

Authors’ reply to Reviewer’s comment 1:

The text has been revised as suggested by the Reviewer.

Reviewer’s comment 3:

Methods 2.11. Was the open arm maze (and other apparatus) were cleaned between tests to eliminate residual odours.

Authors’ reply to Reviewer’s comment 3:

We thank the Reviewer for this comment. Immediately after each session, the apparatus was thoroughly cleaned with cotton pads wetted with 70% ethanol–water solution and dried to eliminate residual odors. This information has been added in the revised manuscript.

Reviewer’s comment 4:

3.4, line 377 and panel 3H. Is “time open entries” correct? (if so what is it?). Shouldn’t the parameters be relative time in open and closed arms, and also number of entries to the open arm?

Authors’ reply to Reviewer’s comment 4:

We thank the Reviewer for pointing this out, and we apologize for providing misleading information. In the revised manuscript, we report the percentage of open arm entries (% Open entries).

Reviewer’s comment 5:

Discussion, line 420. Be careful about linking the behaviour and the cytokine levels as we don’t really know what the relationship (if any) is except they are observed in the same cohorts. You cant really say “behavioural changes were associated with..”. Perhaps something like “Changes in cytokines were also seen in these cohorts” is better.

Authors’ reply to Reviewer’s comment 5:

This point is well taken. We have changed the manuscript as suggested by the Reviewer.

Reviewer’s comment 6:

Lines 435-437. “ We evaluated several experimental protocols described in the literature, assessing the correlation between the severity of the external insult received, the gestational period, the corresponding brain developmental processes”. This should be deleted or rephrased. You didn’t assess severity of injury, gestational insult period or brain development.

Authors’ reply to Reviewer’s comment 6:

As suggested by the Reviewer, this sentence has been removed from the discussion of the revised manuscript.

Reviewer’s comment 7:

Lines 449-451. Have the authors considered separating or excluding the offspring from individual animals that showed a drastic loss of weight. It may be a way to separate sickness from more subtle inflammatory challenge (although of course they are related so may not be sensible?).

Authors’ reply to Reviewer’s comment 7:

We thank the Reviewers for pointing this out. In our experimental cohorts, we found that weight reduction was a characteristic of the entire group of LPS-treated dams, in line with the evidence that administration of LPS results in sickness behaviors, followed by full resolution in 6–18 h depending on the dose (Galic et al., 2012, PMID: 22214786; Clark et al., 2015, PMID: 25257108; Kirsten et al., 2012, PMID: 22714803).

Reviewer’s comment 8:

Line 469 Given the reduced USVs observed, is it really “normal….cognitive processing”?

Authors’ reply to Reviewer’s comment 8:

We agree with the Reviewer and we have rephrased this sentence accordingly.

Reviewer’s comment 9:

Lines 523-527 discusses IFN changes and how it impacts fetal development but you havnt measured fetal IFN, so parts of this not relevant. More relevant is whether adult increases in IFN (as you saw) cause cognitive abnormalities – eg could describe acute effects – or whether adult IFN changes reflect fetal changes.

Authors’ reply to Reviewer’s comment 9:

Following the Reviewer’s comment, we have rephrased this sentence.  

Reviewer’s comment 10:

Lines 552-553. Very good point.

Authors’ reply to Reviewer’s comment 10:

We are very pleased that the Reviewer agree with our interpretation.

Reviewer 3 Report

This manuscript describes the role of maternal immune activation due to LPS stimulation leading to autistic like behaviors. The manuscript is well written and the methods are well planned. I have a few suggestions for this manuscript.

1) Have you looked at sex differences during the cytokine assays? There have been several reports now in the recent years that sex differences can also cause inflammation differences. Have you seen that in your model system?

2) While I do understand that it may be hard to analyse, but it seems that the sampling method for cytokine analysis is from the blood, therefore the cytokines may be peripheral and not brain induced. If they are peripheral cytokines, do the authors know how these are affecting the brain?

Author Response

Reviewer #3

Reviewer #3 (Comments to the Authors)

This manuscript describes the role of maternal immune activation due to LPS stimulation leading to autistic like behaviors. The manuscript is well written and the methods are well planned. I have a few suggestions for this manuscript.

Reviewer’s comment 1:

Have you looked at sex differences during the cytokine assays? There have been several reports now in the recent years that sex differences can also cause inflammation differences. Have you seen that in your model system?

Authors’ reply to Reviewer’s comment 1:

We thank the Reviewer for this comment, and we completely agree that the female offspring should be considered in a follow-up study. In this study, we performed cytokine analysis (and behavioral characterization) only in the male offspring. There is evidence showing that the male offspring has higher susceptibility to intra-uterine stress, compared to females, due to their faster microglial maturation pathway, which is connected with differential gene expression of immune-related genes (Ganguli and Chavali 2021, PMID: 34858132; Hanamsagar et al. 2017, PMID: 28618077; Werling  2016, PMID: 27891212; Werling and Geschwind 2013, PMID: 23406909). Thus, male offspring’s immunological homeostasis is particularly affected by maternal immune activation. However, since sex-dependent differences in preclinical models of ASD have been documented, we will consider the inclusion of both male and female animals in a follow-up study. This has been now briefly discussed in the manuscript.

Reviewer’s comment 2:

While I do understand that it may be hard to analyse, but it seems that the sampling method for cytokine analysis is from the blood, therefore the cytokines may be peripheral and not brain induced. If they are peripheral cytokines, do the authors know how these are affecting the brain?

Authors’ reply to Reviewer’s comment 2:

This is a very good point. There is evidence that peripheral inflammation (as that induced by LPS) causes a “mirror” inflammatory response in the CNS, characterized by additional synthesis and action of cytokines within the brain (Konsman et al., 2022 PMID: 35215252). Although the precise mechanisms responsible for CNS synthesis of cytokines are not entirely understood, the cytokines ultimately produced in the brain can be from several sources including microglia, invading inflammatory cells, microvessel endothelial cells, astrocytes and even neurons where cytokines can be constitutively expressed (Galic et al., 2012 PMID: 22214786). Thus, investigating how and when the peripheral cytokines affect the brain remains an intriguing issue to be investigated. This issue is now mentioned in the discussion of the revised manuscript.
